# MCMC for Variationally Sparse Gaussian Processes

**James Hensman**
CHICAS, Lancaster University
james.hensman@lancaster.ac.uk

**Alexander G. de G. Matthews**
University of Cambridge
am554@cam.ac.uk

**Maurizio Filippone**
EURECOM
maurizio.filippone@eurecom.fr

**Zoubin Ghahramani**
University of Cambridge
zoubin@cam.ac.uk

## Abstract

Gaussian process (GP) models form a core part of probabilistic machine learning. Considerable research effort has been made into attacking three issues with GP models: how to compute efficiently when the number of data is large; how to approximate the posterior when the likelihood is not Gaussian and how to estimate covariance function parameter posteriors. This paper simultaneously addresses these, using a variational approximation to the posterior which is sparse in support of the function but otherwise free-form. The result is a Hybrid Monte-Carlo sampling scheme which allows for a non-Gaussian approximation over the function values and covariance parameters simultaneously, with efficient computations based on inducing-point sparse GPs. Code to replicate each experiment in this paper is available at github.com/sparseMCMC.

## 1 Introduction

Gaussian process models are attractive for machine learning because of their flexible nonparametric nature. By combining a GP prior with different likelihoods, a multitude of machine learning tasks can be tackled in a probabilistic fashion [1]. There are three things to consider when using a GP model: approximation of the posterior function (especially if the likelihood is non-Gaussian), computation, storage and inversion of the covariance matrix, which scales poorly in the number of data; and estimation (or marginalization) of the covariance function parameters. A multitude of approximation schemes have been proposed for efficient computation when the number of data is large. Early strategies were based on retaining a sub-set of the data [2]. Snelson and Ghahramani [3] introduced an inducing point approach, where the model is augmented with additional variables, and Titsias [4] used these ideas in a variational approach. Other authors have introduced approximations based on the spectrum of the GP [5, 6], or which exploit specific structures within the covariance matrix [7, 8], or by making unbiased stochastic estimates of key computations [9]. In this work, we extend the variational inducing point framework, which we prefer for general applicability (no specific requirements are made of the data or covariance function), and because the variational inducing point approach can be shown to minimize the KL divergence to the posterior process [10].

To approximate the posterior function and covariance parameters, Markov chain Monte-Carlo (MCMC) approaches provide asymptotically exact approximations. Murray and Adams [11] and Filippone et al. [12] examine schemes which iteratively sample the function values and covariance parameters. Such sampling schemes require computation and inversion of the full covariance matrix at each iteration, making them unsuitable for large problems. Computation may be reduced somewhat by considering variational methods, approximating the posterior using some fixed family of distributions [13, 14, 15, 16, 1, 17], though many covariance matrix inversions are generally required. Recent works [18, 19, 20] have proposed inducing point schemes which can reduce the

Table 1: Existing variational approaches

| Reference | $p(\mathbf{y} \mid \mathbf{f})$ | Sparse | Posterior | Hyperparam. |
|---|---|---|---|---|
| Williams & Barber[21] [also 14, 17] | probit/logit | ✗ | Gaussian (assumed) | point estimate |
| Titsias [4] | Gaussian | ✓ | Gaussian (optimal) | point estimate |
| Chai [18] | softmax | ✓ | Gaussian (assumed) | point estimate |
| Nguyen and Bonilla [1] | any factorized | ✗ | Mixture of Gaussians | point estimate |
| Hensman et al. [20] | probit | ✓ | Gaussian (assumed) | point estimate |
| This work | any factorized | ✓ | free-form | free-form |

computation required substantially, though the posterior is assumed Gaussian and the covariance parameters are estimated by (approximate) maximum likelihood. Table 1 places our work in the context of existing variational methods for GPs.

This paper presents a general inference scheme, with the only concession to approximation being the variational inducing point assumption. Non-Gaussian posteriors are permitted through MCMC, with the computational benefits of the inducing point framework. The scheme jointly samples the inducing-point representation of the function with the covariance function parameters; with sufficient inducing points our method approaches full Bayesian inference over GP values and the covariance parameters. We show empirically that the number of required inducing points is substantially smaller than the dataset size for several real problems.

## 2   Stochastic process posteriors

The model is set up as follows. We are presented with some data inputs $\mathbf{X} = \{\mathbf{x}_n\}_{n=1}^{N}$ and responses $\mathbf{y} = \{y_n\}_{n=1}^{N}$. A latent function is assumed drawn from a GP with zero mean and covariance function $k(\mathbf{x}, \mathbf{x}')$ with (hyper-) parameters $\boldsymbol{\theta}$. Consistency of the GP means that only those points with data are considered: the latent vector $\mathbf{f}$ represents the values of the function at the observed points $\mathbf{f} = \{f(\mathbf{x}_n)\}_{n=1}^{N}$, and has conditional distribution $p(\mathbf{f} \mid \mathbf{X}, \boldsymbol{\theta}) = \mathcal{N}(\mathbf{f} \mid \mathbf{0}, \mathbf{K}_{ff})$, where $\mathbf{K}_{ff}$ is a matrix composed of evaluating the covariance function at all pairs of points in $\mathbf{X}$. The data likelihood depends on the latent function values: $p(\mathbf{y} \mid \mathbf{f})$. To make a prediction for latent function value test points $\mathbf{f}^\star = \{f(\mathbf{x}^\star)\}_{\mathbf{x}^\star \in \mathbf{X}^\star}$, the posterior function values and parameters are integrated:

$$p(\mathbf{f}^\star \mid \mathbf{y}) = \int\int p(\mathbf{f}^\star \mid \mathbf{f}, \boldsymbol{\theta}) p(\mathbf{f}, \boldsymbol{\theta} \mid \mathbf{y}) \, \mathrm{d}\boldsymbol{\theta} \, \mathrm{d}\mathbf{f} \,. \tag{1}$$

In order to make use of the computational savings offered by the variational inducing point framework [4], we introduce additional input points to the function $\mathbf{Z}$ and collect the responses of the function at that point into the vector $\mathbf{u} = \{\mathbf{u}_m = f(\mathbf{z}_m)\}_{m=1}^{M}$. With some variational posterior $q(\mathbf{u}, \boldsymbol{\theta})$, new points are predicted similarly to the exact solution

$$q(\mathbf{f}^\star) = \int\int p(\mathbf{f}^\star \mid \mathbf{u}, \boldsymbol{\theta}) q(\mathbf{u}, \boldsymbol{\theta}) \, \mathrm{d}\boldsymbol{\theta} \, \mathrm{d}\mathbf{u} \,. \tag{2}$$

This makes clear that the approximation is a stochastic process in the same fashion as the true posterior: the length of the predictions vector $\mathbf{f}^\star$ is potentially unbounded, covering the whole domain.

To obtain a variational objective, first consider the support of $\mathbf{u}$ under the true posterior, and for $\mathbf{f}$ under the approximation. In the above, these points are subsumed into the prediction vector $\mathbf{f}^\star$: from here we shall be more explicit, letting $\mathbf{f}$ be the points of the process at $\mathbf{X}$, $\mathbf{u}$ be the points of the process at $\mathbf{Z}$ and $\mathbf{f}^\star$ be a large vector containing all other points of interest[1]. All of the free parameters of the model are then $\mathbf{f}^\star, \mathbf{f}, \mathbf{u}, \boldsymbol{\theta}$, and using a variational framework, we aim to minimize the Kullback-Leibler divergence between the approximate and true posteriors:

$$\mathcal{K} \triangleq \mathrm{KL}[q(\mathbf{f}^\star, \mathbf{f}, \mathbf{u}, \boldsymbol{\theta}) || p(\mathbf{f}^\star, \mathbf{f}, \mathbf{u}, \boldsymbol{\theta} \mid \mathbf{y})] = \underset{q(\mathbf{f}^\star, \mathbf{f}, \mathbf{u}, \boldsymbol{\theta})}{-\mathbb{E}} \left[ \log \frac{p(\mathbf{f}^\star \mid \mathbf{u}, \mathbf{f}, \boldsymbol{\theta}) p(\mathbf{u} \mid \mathbf{f}, \boldsymbol{\theta}) p(\mathbf{f}, \boldsymbol{\theta} \mid \mathbf{y})}{p(\mathbf{f}^\star \mid \mathbf{u}, \mathbf{f}, \boldsymbol{\theta}) p(\mathbf{f} \mid \mathbf{u}, \boldsymbol{\theta}) q(\mathbf{u}, \boldsymbol{\theta})} \right] \tag{3}$$

where the conditional distributions for $\mathbf{f}^\star$ have been expanded to make clear that they are the same under the true and approximate posteriors, and $\mathbf{X}, \mathbf{Z}$ and $\mathbf{X}^\star$ have been omitted for clarity. Straightforward identities simplify the expression,

$$
\begin{aligned}
\mathcal{K} &= -\mathbb{E}_{q(\mathbf{f},\mathbf{u},\boldsymbol{\theta})}\left[\log \frac{p(\mathbf{u}\,|\,\mathbf{f},\boldsymbol{\theta})p(\mathbf{f}\,|\,\boldsymbol{\theta})p(\boldsymbol{\theta})p(\mathbf{y}\,|\,\mathbf{f})/p(\mathbf{y})}{p(\mathbf{f}\,|\,\mathbf{u},\boldsymbol{\theta})q(\mathbf{u},\boldsymbol{\theta})}\right] \\
&= -\mathbb{E}_{q(\mathbf{f},\mathbf{u},\boldsymbol{\theta})}\left[\log \frac{p(\mathbf{u}\,|\,\boldsymbol{\theta})p(\boldsymbol{\theta})p(\mathbf{y}\,|\,\mathbf{f})}{q(\mathbf{u},\boldsymbol{\theta})}\right] + \log p(\mathbf{y}),
\end{aligned}
\tag{4}
$$

resulting in the variational inducing-point objective investigated by Titsias [4], aside from the inclusion of $\boldsymbol{\theta}$. This can be rearranged to give the following informative expression

$$
\mathcal{K} = \mathrm{KL}\left[q(\mathbf{u},\boldsymbol{\theta})||\frac{p(\mathbf{u}\,|\,\boldsymbol{\theta})p(\boldsymbol{\theta})\exp\{\mathbb{E}_{p(\mathbf{f}\,|\,\mathbf{u},\boldsymbol{\theta})}[\log p(\mathbf{y}\,|\,\mathbf{f})]\}}{C}\right] - \log C + \log p(\mathbf{y}).
\tag{5}
$$

Here $C$ is an intractable constant which normalizes the distribution and is independent of $q$. Minimizing the KL divergence on the right hand side reveals that the optimal variational distribution is

$$
\log \hat{q}(\mathbf{u},\boldsymbol{\theta}) = \mathbb{E}_{p(\mathbf{f}\,|\,\mathbf{u},\boldsymbol{\theta})}\left[\log p(\mathbf{y}\,|\,\mathbf{f})\right] + \log p(\mathbf{u}\,|\,\boldsymbol{\theta}) + \log p(\boldsymbol{\theta}) - \log C.
\tag{6}
$$

For general likelihoods, since the optimal distribution does not take any particular form, we intend to sample from it using MCMC, thus combining the benefits of variationally-sparse Gaussian processes with a free-form posterior. Sampling is feasible using standard methods since $\log \hat{q}$ is computable up to a constant, using $\mathcal{O}(NM^2)$ computations. After completing this work, it was brought to our attention that a similar suggestion had been made in [22], though the idea was dismissed because "prediction in sparse GP models typically involves some additional approximations". Our presentation of the approximation consisting of the entire stochastic process makes clear that no additional approximations are required. To sample effectively, the following are proposed.

**Whitening the prior**  Noting that the problem (6) appears similar to a standard GP for $\mathbf{u}$, albeit with an interesting 'likelihood', we make use of an ancillary augmentation $\mathbf{u} = \mathbf{R}\mathbf{v}$, with $\mathbf{R}\mathbf{R}^\top = \mathbf{K}_{uu}$, $\mathbf{v} \sim \mathcal{N}(\mathbf{0},\mathbf{I})$. This results in the optimal variational distribution

$$
\log \hat{q}(\mathbf{v},\boldsymbol{\theta}) = \mathbb{E}_{p(\mathbf{f}\,|\,\mathbf{u}=\mathbf{R}\mathbf{v})}\left[\log p(\mathbf{y}\,|\,\mathbf{f})\right] + \log p(\mathbf{v}) + \log p(\boldsymbol{\theta}) - \log C
\tag{7}
$$

Previously [11, 12] this parameterization has been used with schemes which alternate between sampling the latent function values (represented by $\mathbf{v}$ or $\mathbf{u}$) and the parameters $\boldsymbol{\theta}$. Our scheme uses HMC across $\mathbf{v}$ and $\boldsymbol{\theta}$ jointly, whose effectiveness is examined throughout the experiment section.

**Quadrature**  The first term in (6) is the expected log-likelihood. In the case of factorization across the data-function pairs, this results in $N$ one-dimensional integrals. For Gaussian or Poisson likelihood these integrals are tractable, otherwise they can be approximated by Gauss-Hermite quadrature. Given the current sample $\mathbf{v}$, the expectations are computed w.r.t. $p(f_n\,|\,\mathbf{v},\boldsymbol{\theta}) = \mathcal{N}(\mu_n,\gamma_n)$, with:

$$
\boldsymbol{\mu} = \mathbf{A}^\top\mathbf{v}; \;\; \boldsymbol{\gamma} = \mathrm{diag}(\mathbf{K}_{ff} - \mathbf{A}^\top\mathbf{A}); \;\; \mathbf{A} = \mathbf{R}^{-1}\mathbf{K}_{uf}; \;\; \mathbf{R}\mathbf{R}^\top = \mathbf{K}_{uu},
\tag{8}
$$

where the kernel matrices $\mathbf{K}_{uf}, \mathbf{K}_{uu}$ are computed similarly to $\mathbf{K}_{ff}$, but over the pairs in $(\mathbf{X},\mathbf{Z}), (\mathbf{Z},\mathbf{Z})$ respectively. From here, one can compute the expected likelihood and it is subsequently straightforward to compute derivatives in terms of $\mathbf{K}_{uf}, \mathrm{diag}(\mathbf{K}_{ff})$ and $\mathbf{R}$.

**Reverse mode differentiation of Cholesky**  To compute derivatives with respect to $\boldsymbol{\theta}$ and $\mathbf{Z}$ we use reverse-mode differentiation (backpropagation) of the derivative through the Cholesky matrix decomposition, transforming $\partial \log \hat{q}(\mathbf{v},\boldsymbol{\theta})/\partial\mathbf{R}$ into $\partial \log \hat{q}(\mathbf{v},\boldsymbol{\theta})/\partial\mathbf{K}_{uu}$, and then $\partial \log \hat{q}(\mathbf{v},\boldsymbol{\theta})/\partial\boldsymbol{\theta}$. This is discussed by Smith [23], and results in a $\mathcal{O}(M^3)$ operation; an efficient Cython implementation is provided in the supplement.

## 3   Treatment of inducing point positions & inference strategy

A natural question is, what strategy should be used to select the inducing points $\mathbf{Z}$? In the original inducing point formulation [3], the positions $\mathbf{Z}$ were treated as parameters to be optimized. One could interpret them as parameters of the *approximate prior covariance* [24]. The variational formulation

[4] treats them as parameters of the variational approximation, thus protecting from over-fitting as they form part of the variational posterior. In this work, since we propose a Bayesian treatment of the model, we question whether it is feasible to treat $\mathbf{Z}$ in a Bayesian fashion.

Since $\mathbf{u}$ and $\mathbf{Z}$ are auxiliary parameters, the form of their distribution does not affect the marginals of the model. The term $p(\mathbf{u}\,|\,\mathbf{Z})$ has been defined by the consistency with the GP in order to preserve the posterior-process interpretation above (i.e. $\mathbf{u}$ should be points on the GP), but we are free to choose $p(\mathbf{Z})$. Omitting dependence on $\boldsymbol{\theta}$ for clarity, and choosing w.l.o.g. $q(\mathbf{u},\mathbf{Z}) = q(\mathbf{u}\,|\,\mathbf{Z})q(\mathbf{Z})$, the bound on the marginal likelihood, similarly to (4) is given by

$$\mathcal{L} = \mathbb{E}_{p(\mathbf{f}\,|\,\mathbf{u},\mathbf{Z})q(\mathbf{u}\,|\,\mathbf{Z})q(\mathbf{Z})}\left[\log\frac{p(\mathbf{y}\,|\,\mathbf{f})p(\mathbf{u}\,|\,\mathbf{Z})p(\mathbf{Z})}{q(\mathbf{u}\,|\,\mathbf{Z})q(\mathbf{Z})}\right]. \tag{9}$$

The bound can be maximized w.r.t $p(\mathbf{Z})$ by noting that the term only appears inside a (negative) KL divergence: $-\mathbb{E}_{q(\mathbf{Z})}[\log q(\mathbf{Z})/p(\mathbf{Z})]$. Substituting the optimal $p(\mathbf{Z}) = q(\mathbf{Z})$ reduces (9) to

$$\mathcal{L} = \mathbb{E}_{q(\mathbf{Z})}\left[\mathbb{E}_{p(\mathbf{f}\,|\,\mathbf{u},\mathbf{Z})q(\mathbf{u}\,|\,\mathbf{Z})}\left[\log\frac{p(\mathbf{y}\,|\,\mathbf{f})p(\mathbf{u}\,|\,\mathbf{Z})}{q(\mathbf{u}\,|\,\mathbf{Z})}\right]\right], \tag{10}$$

which can now be optimized w.r.t. $q(\mathbf{Z})$. Since no entropy term appears for $q(\mathbf{Z})$, the bound is maximized when the distribution becomes a Dirac's delta. In summary, since we are free to choose a prior for $\mathbf{Z}$ which maximizes the amount of information captured by $\mathbf{u}$, the optimal distribution becomes $p(\mathbf{Z}) = q(\mathbf{Z}) = \delta(\mathbf{Z} - \hat{\mathbf{Z}})$. This formally motivates optimizing the inducing points $\mathbf{Z}$.

**Derivatives for $\mathbf{Z}$**    For completeness we also include the derivative of the free form objective with respect to the inducing point positions. Substituting the optimal distribution $\hat{q}(\mathbf{u}, \boldsymbol{\theta})$ into (4) to give $\hat{\mathcal{K}}$ and then differentiating we obtain

$$\frac{\partial\hat{\mathcal{K}}}{\partial\mathbf{Z}} = -\frac{\partial\log C}{\partial\mathbf{Z}} = -\mathbb{E}_{\hat{q}(\mathbf{v},\boldsymbol{\theta})}\left[\frac{\partial}{\partial\mathbf{Z}}\mathbb{E}_{p(\mathbf{f}\,|\,\mathbf{u}=\mathbf{R}\mathbf{v})}\left[\log p(\mathbf{y}\,|\,\mathbf{f})\right]\right]. \tag{11}$$

Since we aim to draw samples from $\hat{q}(\mathbf{v}, \boldsymbol{\theta})$, evaluating this free form inducing point gradient using samples seems plausible but challenging. Instead we use the following strategy.

**1. Fit a Gaussian approximation to the posterior.** We follow [20] in fitting a Gaussian approximation to the posterior. The positions of the inducing points are initialized using k-means clustering of the data. The values of the latent function are represented by a mean vector (initialized randomly) and a lower-triangular matrix $\mathbf{L}$ forms the approximate posterior covariance as $\mathbf{L}\mathbf{L}^{\top}$. For large problems (such as the MNIST experiment), stochastic optimization using AdaDelta is used. Otherwise, LBFGS is used. After a few hundred iterations with the inducing points positions fixed, they are optimized in free-form alongside the variational parameters and covariance function parameters.

**2. Initialize the model using the approximation.** Having found a satisfactory approximation, the HMC strategy takes the optimized inducing point positions from the Gaussian approximation. The initial value of $\mathbf{v}$ is drawn from the Gaussian approximation, and the covariance parameters are initialized at the (approximate) MAP value.

**3. Tuning HMC.** The HMC algorithm has two free parameters to tune, the number of leapfrog steps and the step-length. We follow a strategy inspired by Wang et al. [25], where the number of leapfrog steps is drawn randomly from 1 to $L_{max}$, and Bayesian optimization is used to maximize the expected square jump distance (ESJD), penalized by $\sqrt{L_{max}}$. Rather than allow an adaptive (but convergent) scheme as [25], we run the optimization for 30 iterations of 30 samples each, and use the best parameters for a long run of HMC.

**4. Run tuned HMC to obtain predictions.** Having tuned the HMC, it is run for several thousand iterations to obtain a good approximation to $\hat{q}(\mathbf{v}, \boldsymbol{\theta})$. The samples are used to estimate the integral in equation (2). The following section investigates the effectiveness of the proposed sampling scheme.

## 4   Experiments

### 4.1   Efficient sampling using Hamiltonian Monte Carlo

This section illustrates the effectiveness of Hamiltonian Monte Carlo in sampling from $\hat{q}(\mathbf{v}, \boldsymbol{\theta})$. As already pointed out, the form assumed by the optimal variational distribution $\hat{q}(\mathbf{v}, \boldsymbol{\theta})$ in equation (6) resembles the joint distribution in a GP model with a non-Gaussian likelihood.

For a fixed $\boldsymbol{\theta}$, sampling $\mathbf{v}$ is relatively straightforward, and this can be done efficiently using HMC [12, 26, 27] or Elliptical Slice Sampling [28]. A well tuned HMC has been reported to be extremely efficient in sampling the latent variables, and this motivates our effort into trying to extend this efficiency to the sampling of hyper-parameters as well. This is also particularly appealing due to the convenience offered by the proposed representation of the model.

The problem of drawing samples from the posterior distribution over $\mathbf{v}, \boldsymbol{\theta}$ has been investigated in detail in [11, 12]. In these works, it has been advocated to alternate between the sampling of $\mathbf{v}$ and $\boldsymbol{\theta}$ in a Gibbs sampling fashion and condition the sampling of $\boldsymbol{\theta}$ on a suitably chosen transformation of the latent variables. For each likelihood model, we compare efficiency and convergence speed of the proposed HMC sampler with a Gibbs sampler where $\mathbf{v}$ is sampled using HMC and $\boldsymbol{\theta}$ is sampled using the Metropolis-Hastings algorithm. To make the comparison fair, we imposed the mass matrix in HMC and the covariance in MH to be isotropic, and any parameters of the proposal were tuned using Bayesian optimization. Unlike in the proposed HMC sampler, for the Gibbs sampler we did not penalize the objective function of the Bayesian optimization for large numbers of leapfrog steps, as in this case HMC proposals on the latent variables are computationally cheaper than those on the hyper-parameters. We report efficiency in sampling from $\hat{q}(\mathbf{v}, \boldsymbol{\theta})$ using Effective Sample Size (ESS) and Time Normalized (TN)-ESS. In the supplement we include convergence plots based on the Potential Scale Reduction Factor (PSRF) computed based on ten parallel chains; in these each chain is initialized from the VB solution and individually tuned using Bayesian optimization.

## 4.2   Binary Classification

We first use the *image* dataset [29] to investigate the benefits of the approach over a Gaussian approximation, and to investigate the effect of changing the number of inducing points, as well as optimizing the inducing points under the Gaussian approximation. The data are 18 dimensional: we investigated the effect of our approximation using both ARD (one lengthscale per dimension) and an isotropic RBF kernel. The data were split randomly into 1000/1019 train/test sets; the log predictive density over ten random splits is shown in Figure 1.

Following the strategy outlined above, we fitted a Gaussian approximation to the posterior, with $\mathbf{Z}$ initialized with k-means. Figure 1 investigates the difference in performance when $\mathbf{Z}$ is optimized using the Gaussian approximation, compared to just using k-means for $\mathbf{Z}$. Whilst our strategy is not guaranteed to find the global optimum, it is clear that it improves the performance.

The second part of Figure 1 shows the performance improvement of our sampling approach over the Gaussian approximation. We drew 10,000 samples, discarding the first 1000: we see a consistent improvement in performance once $M$ is large enough. For small $M$, The Gaussian approximation appears to work very well. The supplement contains a similar Figure for the case where a single lengthscale is shared: there, the improvement of the MCMC method over the Gaussian approximation is smaller but consistent. We speculate that the larger gains for ARD are due to posterior uncertainty in the lengthscales, which is poorly represented by a point in the Gaussian/MAP approximation.

The ESS and TN-ESS are comparable between HMC and the Gibbs sampler. In particular, for 100 inducing points and the RBF covariance, ESS and TN-ESS for HMC are 11 and $1.0 \cdot 10^{-3}$ and for the Gibbs sampler are 53 and $5.1 \cdot 10^{-3}$. For the ARD covariance, ESS and TN-ESS for HMC are 14 and $5.1 \cdot 10^{-3}$ and for the Gibbs sampler are 1.6 and $1.5 \cdot 10^{-4}$. Convergence, however, seems to be faster for HMC, especially for the ARD covariance (see the supplement).

## 4.3   Log Gaussian Cox Processes

We apply our methods to Log Gaussian Cox processes [30]: doubly stochastic models where the rate of an inhomogeneous Poisson process is given by a Gaussian process. The main difficulty for inference lies in that the likelihood of the GP requires an integral over the domain, which is typically intractable. For low dimensional problems, this integral can be approximated on a grid; assuming that the GP is constant over the width of the grid leads to a factorizing Poisson likelihood for each of the grid points. Whilst some recent approaches allow for a grid-free approach [19], these usually require concessions in the model, such as an alternative link function, and do not approach full Bayesian inference over the covariance function parameters.

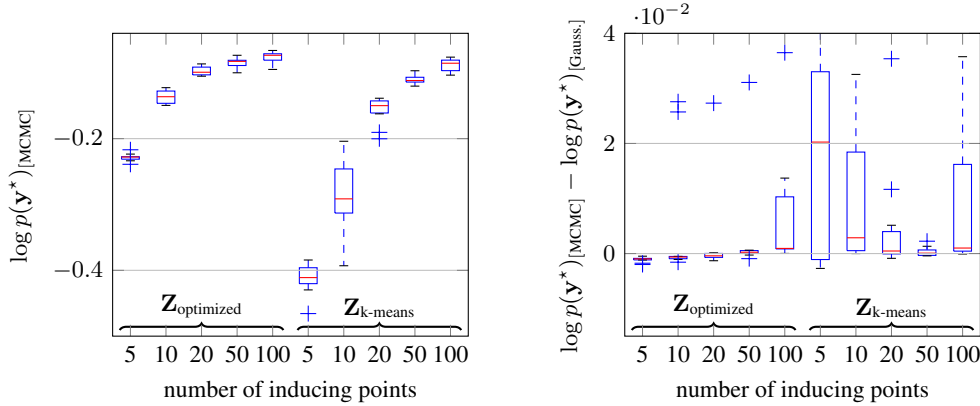

Figure 1: Performance of the method on the *image* dataset, with one lengthscale per dimension. Left, box-plots show performance for varying numbers of inducing points and **Z** strategies. Optimizing **Z** using the Gaussian approximation offers significant improvement over the k-means strategy. Right: improvement of the performance of the Gaussian approximation method, with the same inducing points. The method offers consistent performance gains when the number of inducing points is larger. The supplement contains a similar figure with only a single lengthscale.

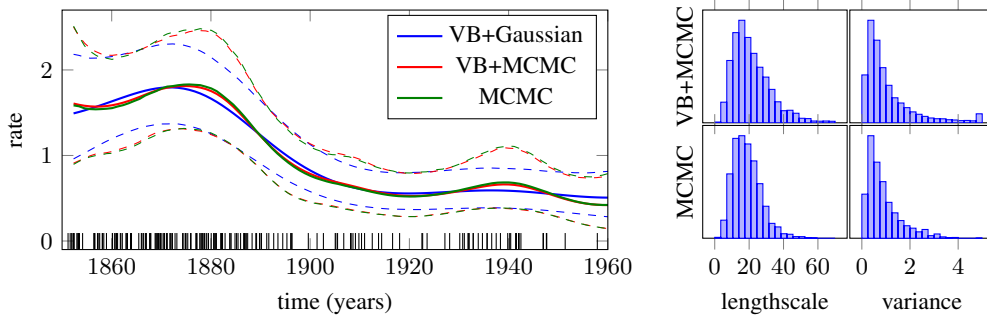

Figure 2: The posterior of the rates for the coal mining disaster data. Left: posterior rates using our variational MCMC method and a Gaussian approximation. Data are shown as vertical bars. Right: posterior samples for the covariance function parameters using MCMC. The Gaussian approximation estimated the parameters as $(12.06, 0.55)$.

**Coal mining disasters**  On the one-dimensional coal-mining disaster data. We held out 50% of the data at random, and using a grid of 100 points with 30 evenly spaced inducing points **Z**, fitted both a Gaussian approximation to the posterior process with an (approximate) MAP estimate for the covariance function parameters (variance and lengthscale of an RBF kernel). With Gamma priors on the covariance parameters we ran our sampling scheme using HMC, drawing 3000 samples. The resulting posterior approximations are shown in Figure 2, alongside the true posterior using a sampling scheme similar to ours (but without the inducing point approximation). The free-form variational approximation matches the true posterior closely, whilst the Gaussian approximation misses important detail. The approximate and true posteriors over covariance function parameters are shown in the right hand part of Figure 2, there is minimal discrepancy in the distributions.

Over 10 random splits of the data, the average held-out log-likelihood was $-1.229$ for the Gaussian approximation and $-1.225$ for the free-form MCMC variant; the average difference was $0.003$, and the MCMC variant was always better than the Gaussian approximation. We attribute this improved performance to marginalization of the covariance function parameters.

Efficiency of HMC is greater than for the Gibbs sampler; ESS and TN-ESS for HMC are $6.7$ and $3.1 \cdot 10^{-2}$ and for the Gibbs sampler are $9.7$ and $1.9 \cdot 10^{-2}$. Also, chains converge within few thousand iterations for both methods, although convergence for HMC is faster (see the supplement).

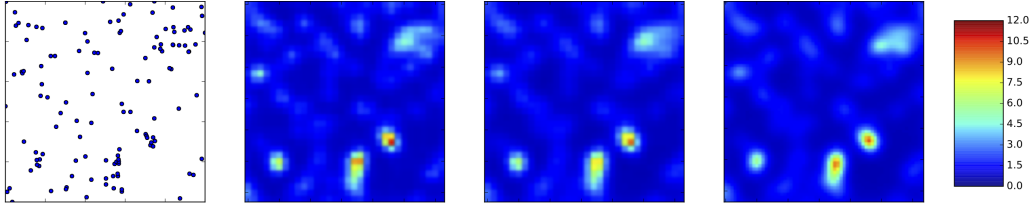

Figure 3: Pine sapling data. From left to right: reported locations of pine saplings; posterior mean intensity on a 32x32 grid using full MCMC; posterior mean intensity on a 32x32 grid (with sparsity using 225 inducing points), posterior mean intensity on a 64x64 grid (using 225 inducing points).

**Pine saplings**    The advantages of the proposed approximation are prominent as the number of grid points become higher, an effect emphasized with increasing dimension of the domain. We fitted a similar model to the above to the pine sapling data [30].

We compared the sampling solution obtained using 225 inducing points on a 32 x 32 grid to the gold standard full MCMC run with the same prior and grid size. Figure 3 shows that the agreement between the variational sampling and full sampling is very close. However the variational method was considerably faster. Using a single core on a desktop computer required 3.4 seconds to obtain 1 effective sample for a well tuned variational method whereas it took 554 seconds for well tuned full MCMC. This effect becomes even larger as we increase the resolution of the grid to 64 x 64, which gives a better approximation to the underlying smooth function as can be seen in figure 3. It took 4.7 seconds to obtain one effective sample for the variational method, but now gold standard MCMC comparison was computationally extremely challenging to run for even a single HMC step. This is because it requires linear algebra operations using $\mathcal{O}(N^3)$ flops with $N = 4096$.

## 4.4    Multi-class Classification

To do multi-class classification with Gaussian processes, one latent function is defined for each of the classes. The functions are defined a-priori independent, but covary a posteriori because of the likelihood. Chai [18] studies a sparse variational approximation to the softmax multi-class likelihood restricted to a Gaussian approximation. Here, following [31, 32, 33], we use a robust-max likelihood. Given a vector $\mathbf{f}_n$ containing $K$ latent functions evaluated at the point $\mathbf{x}_n$, the probability that the label takes the integer value $y_n$ is $1 - \epsilon$ if $y_n = \text{argmax} \, \mathbf{f}_n$ and $\epsilon/K - 1$ otherwise. As Girolami and Rogers [31] discuss, the 'soft' probit-like behaviour is recovered by adding a diagonal 'nugget' to the covariance function. In this work, $\epsilon$ was fixed to $0.001$, though it would also be possible to treat this as a parameter for inference. The expected log-likelihood is $\mathbb{E}_{p(\mathbf{f}_n \, | \, \mathbf{v}, \boldsymbol{\theta})}[\log p(y_n \, | \, \mathbf{f}_n)] = p \log(\epsilon) + (1-p) \log(\epsilon/(K-1))$, where $p$ is the probability that the labelled function is largest, which is computable using one-dimensional quadrature. An efficient Cython implementation is contained in the supplement.

**Toy example**    To investigate the proposed posterior approximation for the multivariate classification case, we turn to the toy data shown in Figure 4. We drew 750 data points from three Gaussian distributions. The synthetic data was chosen to include non-linear decision boundaries and ambiguous decision areas. Figure 4 shows that there are differences between the variational and sampling solutions, with the sampling solution being more conservative in general (the contours of 95% confidence are smaller). As one would expect at the decision boundary there are strong correlations between the functions which could not be captured by the Gaussian approximation we are using. Note the movement of inducing points away from k-means and towards the decision boundaries.

Efficiency of HMC and the Gibbs sampler is comparable. In the RBF case, ESS and TN-ESS for HMC are 1.9 and $3.8 \cdot 10^{-4}$ and for the Gibbs sampler are 2.5 and $3.6 \cdot 10^{-4}$. In the ARD case, ESS and TN-ESS for HMC are 1.2 and $2.8 \cdot 10^{-3}$ and for the Gibbs sampler are 5.1 and $6.8 \cdot 10^{-4}$. For both cases, the Gibbs sampler struggles to reach convergence even though the average acceptance rates are similar to those recommended for the two samplers individually.

**MNIST**    The MNIST dataset is a well studied benchmark with a defined training/test split. We used 500 inducing points, initialized from the training data using k-means. A Gaussian approximation

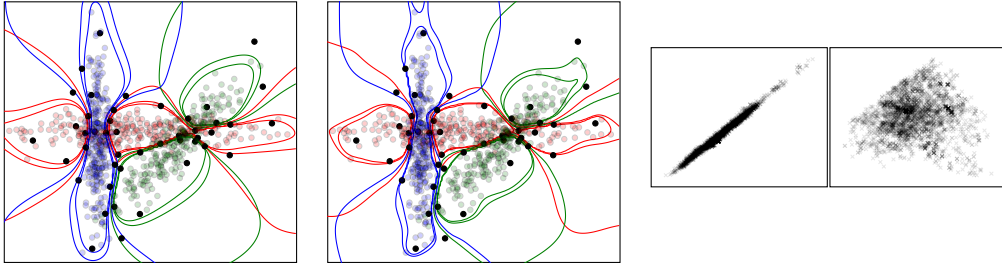

Figure 4: A toy multiclass problem. Left: the Gaussian approximation, colored points show the simulated data, lines show posterior probability contours at 0.3, 0.95, 0.99. Inducing points positions shows as black points. Middle: the free form solution with 10,000 posterior samples. The free-form solution is more conservative (the contours are smaller). Right: posterior samples for $\mathbf{v}$ at the same position but across different latent functions. The posterior exhibits strong correlations and edges.

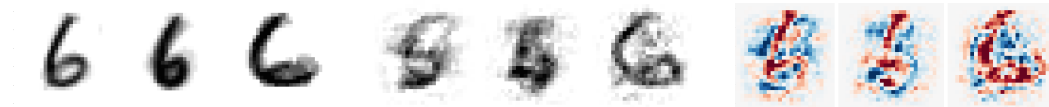

Figure 5: Left: three k-means centers used to initialize the inducing point positions. Center: the positions of the same inducing points after optimization. Right: difference.

was optimized using minibatch-based optimization over the means and variances of $q(\mathbf{u})$, as well as the inducing points and covariance function parameters. The accuracy on the held-out data was 98.04%, significantly improving on previous approaches to classify these digits using GP models.

For binary classification, Hensman et al. [20] reported that their Gaussian approximation resulted in movement of the inducing point positions toward the decision boundary. The same effect appears in the multivariate case, as shown in Figure 5, which shows three of the 500 inducing points used in the MNIST problem. The three examples were initialized close to the many six digits, and after optimization have moved close to other digits (five and four). The last example still appears to be a six, but has moved to a more 'unusual' six shape, supporting the function at another extremity. Similar effects are observed for all inducing-point digits. Having optimized the inducing point positions with the approximate $q(\mathbf{v})$, and estimate for $\boldsymbol{\theta}$, we used these optimal inducing points to draw samples from $\mathbf{v}$ and $\boldsymbol{\theta}$. This did not result in an increase in accuracy, but did improve the log-density on the test set from -0.068 to -0.064. Evaluating the gradients for the sampler took approximately 0.4 seconds on a desktop machine, and we were easily able to draw 1000 samples. This dataset size has generally be viewed as challenging in the GP community and consequently there are not many published results to compare with. One recent work [34] reports a 94.05% accuracy using variational inference and a GP latent variable model.

## 5 Discussion

We have presented an inference scheme for general GP models. The scheme significantly reduces the computational cost whilst approaching exact Bayesian inference, making minimal assumptions about the form of the posterior. The improvements in accuracy in comparison with the Gaussian approximation of previous works has been demonstrated, as has the quality of the approximation to the hyper-parameter distribution. Our MCMC scheme was shown to be effective for several likelihoods, and we note that the automatic tuning of the sampling parameters worked well over hundreds of experiments. This paper shows that MCMC methods are feasible for inference in large GP problems, addressing the unfair sterotype of 'slow' MCMC.

**Acknowledgments** JH was funded by an MRC fellowship, AM and ZG by EPSRC grant EP/I036575/1 and a Google Focussed Research award.

## Footnotes

[1] The vector $\mathbf{f}^\star$ here is considered finite but large enough to contain any point of interest for prediction. The infinite case follows Matthews et al. [10], is omitted here for brevity, and results in the same solution.

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
