[Supplementary Material]



# MCMC for Variationally Sparse GPs

## 5.1 Coal data

Figure 6 replicates Figure 1, but with a single lengthscale shared across each input.

Figure 6: Performance of the method on the *image* dataset, with a single lengthscale.

## 5.2 Convergence plots

Convergence of the samplers on the Image dataset is reported in fig. 7 and shows the evolution of the PSRF for the twenty slowest parameters for HMC and the Gibbs sampler in the case of RBF and ARD covariances. The figure shows that HMC consistently converges faster than the Gibbs sampler for both covariances, even when the ESS of the slowest variable is comparable.

Fig. 7 shows the convergence analysis on the coal dataset. In this case, HMC converges faster than the Gibbs sampler and efficiency is comparable.

Convergence of the samplers on the toy multi-class dataset is reported in fig. 9. HMC converges much faster than the Gibbs sampler even though efficiency measured through ESS is comparable.

Figure 7: Image dataset - Evolution of the PSRF of the twenty least efficient parameter traces for HMC (blue) and the Gibbs sampler (red). Left panel: RBF case - minimum ESS and TN-ESS for HMC are 11 and $1.0 \cdot 10^{-3}$ and for the Gibbs sampler are 53 and $5.1 \cdot 10^{-3}$. Right panel: ARD case - minimum ESS and TN-ESS for HMC are 14 and $5.1 \cdot 10^{-3}$ and for the Gibbs sampler are 1.6 and $1.5 \cdot 10^{-4}$.

Figure 8: Coal dataset - Evolution of the PSRF of the twenty least efficient parameter traces for HMC (blue) and the Gibbs sampler (red). Minimum ESS and TN-ESS for HMC are 6.7 and $3.1 \cdot 10^{-2}$ and for the Gibbs sampler are 9.7 and $1.9 \cdot 10^{-2}$.

Figure 9: Multiclass dataset - Evolution of the PSRF of the twenty least efficient parameter traces for HMC (blue) and the Gibbs sampler (red). Left panel: RBF case - minimum ESS and TN-ESS for HMC are 1.9 and $3.8 \cdot 10^{-4}$ and for the Gibbs sampler are 2.5 and $3.6 \cdot 10^{-4}$. Right panel: ARD case - minimum ESS and TN-ESS for HMC are 1.2 and $2.8 \cdot 10^{-3}$ and for the Gibbs sampler are 5.1 and $6.8 \cdot 10^{-4}$.

## 5.3 Pine saplings

Figure 10: A larger version of Figure 3. Top right: gold standard MCMC 32x32 grid. Bottom left: Variational MCMC 32x32 grid. Bottom right: Variational MCMC 64x64 grid, with 225 inducing points in the non-exact case.