[Reviews · NeurIPS 2015]

Submitted by Assigned_Reviewer_1

0. Summary

This paper presents a method for sampling from the (approximate) posterior in variational sparse Gaussian process models with factorizing likelihoods. The main claimed contribution is that allowing a "free-form"

for the approximate posterior, instead of constraining it to belong to a pre-specified

family of distributions.

1. Clarity This is paper is well written with the contributions clearly stated, describing also how the the paper differentiates from previous work.

2. Quality, originality and significance The paper addresses a significant problem in probabilistic inference, that of approximating posterior distributions in models with Gaussian process priors and factorizing likelihoods, allowing efficient sampling of inducing variables and covariance hyper-parameters. The paper is technically sound, building

upon previous work on variational approaches to sparse GP models (reference 5 in the paper) with the main contribution appearing in Equations (5) and (6),

rewriting the variational objective so that one can use MCMC (HMC) machinery to to draw samples from the joint approximate posterior over inducing variables and hyper-parameters. I note in passing that Equation (5) seems to have typo, as the expectation of the conditional likelihood over the conditional prior should be exponentiated (?).

The very same idea of sampling from the approximate posterior within the inducing-point framework has been proposed in the past (see Equation 1.27 and corresponding text in [1] below) but I am unaware of its implementation. I am surprised the authors do not reference this work. Nevertheless, the contribution of the paper is still significant as it carries out the development of a practical framework that does not involve the commonly used assumption of optimizing over the parameters of a fixed distribution, while still maintaining the computational efficiency of sparse GP models.

I do not see major concerns about this paper, although it somehow gives the readers the impression that the proposed method is sampling from the true posterior of a sparse GP model, which I do not think is the case giving the assumed form for the posterior approximation in the denominator of Equation (3). It would be useful if the authors discuss this.

The paper presents results on a wide variety of experiments on small datasets and on MNIST, comparing to the full Gaussian approximation and to 'gold standard' MCMC. It is interesting to see that there is not a large practical difference between a free-form approximation and a Gaussian approximation but I think the proposed approach it is still useful on its own right. On the MNIST data, it would be useful to know how long it took to run the whole experiment.

3. Minor comments - Figures 1 and 4 show some weird numbers '000, 001, ...' - line 370: "As Girolami and S' -> 'As Girolami and Rogers'

- line 401: "approaches to to classify" -> "approaches to classify"

4. References

[1] . M. K. Titsias, M. Rattray and N.D. Lawrence. Markov chain Monte Carlo algorithms for Gaussian processes. Chapter to appear in the book "Inference and Learning in Dynamic Models"
Summary: An interesting paper that introduces MCMC into variational sparse GP models to sample from the approximate variational posterior, instead of optimizing the parameters of a pre-defined family of distributions. The contribution seems significant and the experiments solid throughout.

Submitted by Assigned_Reviewer_2

Quality This contribution seems to address the long standing issue of scaling of GPs in a way that builds on VB and inducing variables. The experimental evaluation seems to be persuasive although it would have been good to have seen a comparison with data sub-sampling to answer the sceptics question of whether a Bayesian practitioner would really need and employ this sort of machinery.

Clarity Nicely presented with no problem in accessing the material content.

Originality this looks to me to be a novel combination of existing methods that seems to work so in this sense I would say it has a fair degree of originality

Signicance This is where the question mark is for me - as it is for most of the approximate inference methodology - if a practitioner in the field will employ this methodology then this is significant work. Will a practitioner use it - I doubt it. The reason being it is not clear how good these approximations really are and what are the degradation and breakdown points. In addition it is not clear how significant this really is when compared to exact sub-sampling. Despite this caveat I think this is pretty good work.
Summary: A variational approximation based on inducing points provides an approximate posterior for a GP with general non-Gaussian likelihood. MCMC is used to sample from this target and a wide range of empirical results are reported which suggest a competitive computational scaling for what would seem to be a minimal loss of accuracy.

Submitted by Assigned_Reviewer_3

Quality and Significance: very good. The theoretical development seems solid and the experiments seem to back it up.

Suggestions that would make it stronger and clear what its significance is: experimental results on a very large, real dataset to give a sense of what scale of data is reasonably possible with this method. I'd also like to know how well your method is able to do hyperparameter learning---complex kernels can be hard to learn with inducing points methods. A synthetic example with a complex kernel might be nice.

Originality: I believe this is novel, but it would be good to clarify how it fits in with other variational approaches (see below).

Clarity: unless I'm missing something, your use of the term free-form simply means that there is no closed form specification for the posterior distribution you learn. This is often the case in with Bayesian models (e.g. a fully Bayesian treatment of a GP, with priors on the hyperparameters, fit with MCMC),

and thus it would be more appropriate to characterize it as a standard (non-conjugate) Bayesian approach, which I would simply describe as learning a distribution over the hyperparameters, as opposed to a point estimate. Free-form makes it sound like you're doing something non-parametric.

You should highlight the novelty of lines 124-126. On first read, I missed this crucial sentence and got very confused by why you were using HMC after going to the trouble of proposing a variational approach. I would definitely like to know more about the reverse mode differentiation of the Cholesky, which I did not see in the supplement.

Now that I think about it, you say you're using variational inference, and you do use KL divergence, but what would have changed in your theory if you'd started with the derivation of FITC and then said you were going to if it with HMC? What's _variational_ about your method?

Tuning HMC---why not use NUTS?

I'm not familiar with PSRT. Can you compute the Gelman-Rubin statistic (Rhat)?

I'm surprised that Figure 2 (left) indicates that the Gaussian approximation has the same degree of uncertainty as the MCMC methods. Shouldn't the greater uncertainty in the parameters vs point estimates lead to more uncertainty in the posterior estimates? How did you calculate these posterior credible intervals?

Figure 2 (right) doesn't have the Gaussian approximation results, but I see them numerically in the caption. I guess you only put them there because they're point estimates? Would be good to thus list the median from MCMC and VB+MCMC for comparison.

Your github repository is empty. ;)
Summary: This paper addresses a very important problem, that of scaling up GP learning and inference for non-Gaussian observation models, using approximate Bayesian methods, while still learning a posterior distribution over the hyperparameters of the covariance function. The approach has an interesting novelty: rather than optimizing a variational objective, MCMC is used to sample a variational distribution.

Author Feedback
Author rebuttal: Reviewer 1

Thanks for your detailed review. You're right, there's a typo in equation 5, sorry! We've included the minor fixes you suggested.

Sorry to have missed the citation, we've added this. We're glad that you agree that the contribution is nevertheless significant in providing a computationally efficient practical framework. We hope you also enjoyed the contribution of interpreting the approximation as a stochastic process.

Regarding "the true posterior of a sparse GP model", this isn't quite the impression we intended to give. We think in terms of a sparse variational approximation to the true posterior process -- the approximation is sparse, not the model. We do choose a form for the approximation as in the denominator of equation 3 as you suggest, and then derive the optimal expression for q(u, \theta) given this form, then sample from this. We've clarified this at the start of section 2.

Fitting the Gaussian approximation takes about 6-8 hours for the 60K training points in MNIST on a desktop. This is in part because of the complexity of computing derivatives wrt the inducing inputs Z in such high dimensions. We usually state the complexity of these methods as O(NM^2), but the derivatives may take O(NM^2 D), where D is the dimensionality of the input space. Running the sampler takes a similar time again, although each iteration is much cheaper because we don't update Z. Future work should cover making the sampling more efficient (stochastic Langevin dynamics, perhaps?)

Reviewer 2

Thanks for your detailed review, we're glad you think the paper is good, and of sufficient quality, clarity and originality.

We're not sure what you refer to by 'exact sub-sampling'. Do you mean the recent work on Unbiased Bayes? Or fitting exact GPs to subsets of the data? In the first case, although this work is very interesting we think there are advantages to obtaining samples from the (approximate) posterior. In the second case, we know that fitting GPs to subsets can often work well, though we only know of results for this in the context of regression. It's difficult to envisage a comprehensive framework covering non-Gaussian likelihoods and hyper-parameter learning using subsets.

Fortunately, although deriving and coding our approximation required significant effort the code will be publicly available in future so we hope that this will aid the uptake of these methods in the community.

Reviewer 3

Thanks for your detailed review, we're pleased you liked the paper. Those are some excellent suggestions for what to do next!

We've not studied a very complex kernel, but we have studied the ability to learn many kernel parameters in an ARD kernel with one parameter (lengthscale) per dimension.

> You should highlight the novelty of lines 124-126
Done!

> What's _variational_ about your method?
Given the form for the approximating family we use, the form of the optimal q(u, \theta) is a necessary consequence of the KL divergence. We've clarified the use of the term 'free form', used as in Beal (2003).

> Why not NUTS?
We had the same thought, and we actually tried NUTS for a while. Tuning HMC turned out to be less arduous than we expected :)

> PSRF
The PSRF is like a Gelman-Rubin statistic per dimension: we then show the 20 worst performing dimensions, as recommended by Gelman.

> Uncertainty in Fig 2.
Uncertainty in variational approximations can be rather subtle as you allude to, but such results are not unheard of. We're confident in the correctness of these results.

> Github repository is empty
Oh. We're sorry you've had difficulty. Did you get the right address? github.com/sparseMCMC/NIPS2015
You'll find reverse mode Cholesky code there.

Reviewer 4: Thanks for your review, we're glad you liked the paper. Note that our variational approximation was chosen to make the least possible assumptions in order to get an O(N M^2) algorithm, and that empirically (error bars in figure 2) we end up with a very good approximation when compared with the slower full MCMC and the fixed form variational approximation.

Reviewer 5: Thanks for your review. Note that our method is not simply HMC on a sparse GP prior - eq (3) shows that our MCMC method minimizes a KL divergence between approximate and posterior stochastic processes. This combination of MCMC and variational approximations is non-trivial and novel in the context of GPs.

Reviewer 6: Thanks for your review. Although many people have explored variational methods and MCMC, it is subtle and non-trivial to combine them successfully in a manner that inherits the benefits of both. This is borne out by our extensive experiments demonstrating scalability (MNIST), applicability to general likelihoods (multi-class, log Gaussian Cox), and accuracy of approximation (Fig 2). Given the strong empirical success we have had we would argue that if it was straightforward to do, someone would have done it sooner :)